# Visible Light Electromagnetic Interaction of PM567 Chiral Dye for Asymmetric Photocatalysis, a First-Principles Investigation

**Yujie Dai [1],[†], Chen Lu [2],[†], Lin Liang [2],[†], Naixing Feng [3],* and Jingang Wang [2],***

1    College of Petroleum Engineering, Liaoing Shihua University, Fushun 113001, China; yujiedai2008@163.com
2    College of Sciences, Liaoning Shihua University, Fushun 113001, China; luchen10288@163.com (C.L.);
     ll1089@126.com (L.L.)
3    Institute of Microscale Optoelectronics, Shenzhen University, Shenzhen 518060, China
*    Correspondence: fengnaixing@szu.edu.cn (N.F.); jingang_wang@lnpu.edu.cn (J.W.);
     Tel.: +86-1804-003-6755 (J.W.)
†    Contributed Equally.

**Abstract:** In asymmetric photocatalytic reactions, it is necessary to study the mechanism of the asymmetric electromagnetic interaction between molecules and light. In this work, we theoretically studied the electromagnetic interactions between the light-induced charge transfer reaction and the chiral reaction of PM567 dye. We found that the chiral responses of molecules in different wavelength ranges were partially due to pyrromethene and binaphthalene. Therefore, the catalytic sites with different chirality also corresponds to the two-part groups. Through quantitative analysis, we found the entire analysis process to be complete and self-consistent.

**Keywords:** PM567; asymmetric photocatalysis; chiral spectrum; ECD; charge transfer

## 1. Introduction

In recent years, the field of photocatalysis [1–5] has rapidly developed. The applications and theoretical research for photocatalysis are endless. As a new catalytic reaction mode, photocatalysis is particularly important for organic chemistry [6], medicinal chemistry [7,8], and natural product synthesis chemistry [9]. Photocatalytic reactions have many advantages: high reaction rate, affordability, and environmentally friendliness [10,11]. Over the years, the means of photocatalytic reactions have been diverse, including surface plasmon enhancement [12], and reactions assisted by two-dimensional materials, such as grapheme and black phosphorus [13,14]. For photocatalytic reactions, the mechanism of photoredox is the core theory. In the photoredox reaction, the catalyst or dye molecule acts as an energy acceptor and converter and an electron provider, and its charge transfer ability at a specific wavelength is significant [15–18] and needs to ensure the stability under the effect of light [19,20].

As a traditional widely-used material, PM567 dye has good charge transfer ability and light stability [21–23]. However, for chiral molecules, the problem of stereoisomerism in asymmetric catalysis needs to be considered. Before this is possible, because of the light-driven catalytic reaction, the chiral electromagnetic interaction when light interacts with molecules must be studied. The chiral electromagnetic interaction between molecules and light is extremely important for asymmetric photocatalytic reactions. The exploration of the chiral electromagnetic interaction mechanism of dyes has guiding significance for the design of asymmetric photocatalytic reactions. In our previous work, we developed a method for electromagnetic transition dipole moment decomposition analysis and achieved certain results in the field of molecular systems and two-dimensional materials [24–26]. Therefore, in this work, we used this

method to theoretically study the photo-induced chiral electromagnetic interaction of PM567 molecules to guide the asymmetric catalytic reaction of photocatalysis.

## 2. Results and Discussion

Figure 1 shows the Lewis and 3D molecular structures of pyrromethene 567 (PM567) dye with binaphthalene. The figure shows that the molecule is composed of two parts connected by boron atoms. Since the connection between the boron atom and the oxygen atom is directional, the pi-conjugated system connected to it is also chiral. It is necessary to study the chirality of this special bonding system because the chirality caused by boron atoms is incredibly special. From the molecular structure, the chirality of PM567 dye is not limited to a certain chiral center, but is determined by a part of the atomic group.

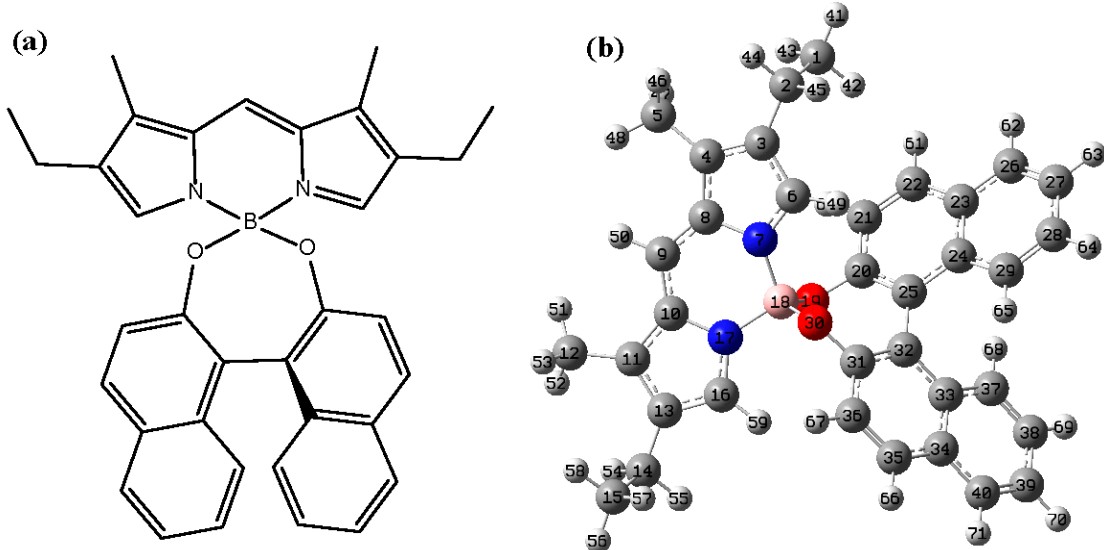

**Figure 1.** The Lewis formula (**a**) and 3D molecular structure (**b**) of PM567 dye.

The electromagnetic interaction caused by this chiral structure is significantly used in visible light chiral photocatalytic (asymmetric photocatalytic) reactions. Therefore, it is necessary to study the nature of chiral electromagnetic interaction in the visible region. For the PM567 dye, the absorption peak dominated by the first excited state is in the visible region (420 nm), see Figure 2. In the near ultraviolet region, the excited states are $S_3$, $S_4$, and $S_6$. The sensitivity of the absorption spectrum is positively related to the vibrator intensity. For the first excited state, longer wavelength light can excite molecules. However, compared to the maximum peak, the absorption intensity is \ and the excitation efficiency are relatively low. In this part of the discussion, we selected these excited states with stronger intensity as the analysis object. Because different excitation states correspond to different excitation energies, the molecular orbits corresponding to the excitation process are also different. Combining the configuration coefficient and molecular orbital during the excitation process, the electron-hole-pair density in the transition process can be calculated and visualized by the following formula:

$$\rho^{hole}(\mathbf{r}) = \rho^{hole}_{(loc)}(\mathbf{r}) + \rho^{hole}_{(cross)}(\mathbf{r}) = \sum_{i \to a} \left(w_i^a\right)^2 \varphi_i(\mathbf{r})\varphi_i(\mathbf{r}) + \sum_{i \to a}\sum_{j \neq i \to a} w_i^a w_j^a \varphi_i(\mathbf{r})\varphi_j(\mathbf{r})$$

$$\rho^{ele}(\mathbf{r}) = \rho^{ele}_{(oc)}(\mathbf{r}) + \rho^{ele}_{(cross)}(\mathbf{r}) = \sum_{i \to a} \left(w_i^a\right)^2 \varphi_a(\mathbf{r})\varphi_a(\mathbf{r}) + \sum_{i \to a i \to b \neq a} w_i^a w_i^b \varphi_a(\mathbf{r})\varphi_b(\mathbf{r})$$

(1)

where the $\varphi$ is the wavefunction of molecular orbital. The subscripts loc and cross represent the contribution of local and cross terms to the electron and hole density. This visualization method has many advantages over molecular orbital analysis methods. First, because molecular orbitals are linear

combinations of atomic orbitals, their wave function forms are dispersed throughout the molecular space, which is inconvenient for the analysis of transition contributions. Second, any excitation process is not limited to a single pair of molecular orbitals, but a combination of excitations between multiple molecular orbitals. Therefore, for the case where the configuration coefficients are evenly distributed, the molecular orbital analysis method is extremely complicated. The electron-hole-pair density can be analyzed with a contour map of the entire excitation process.

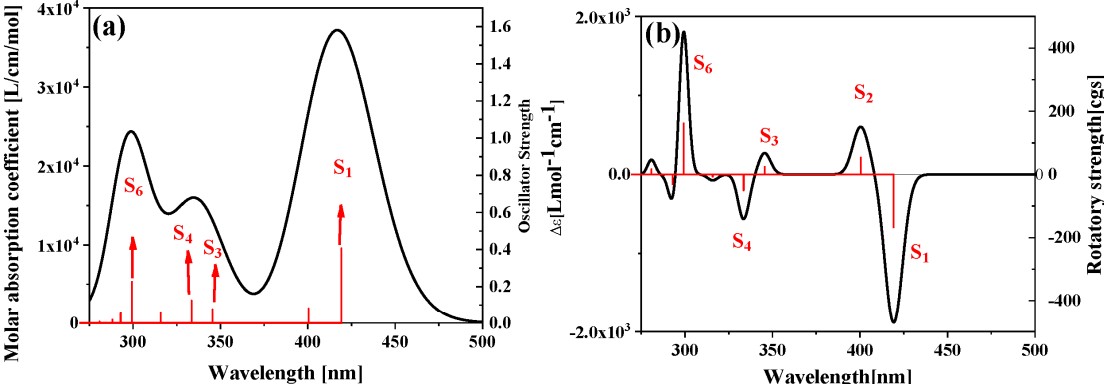

**Figure 2.** The one-photon absorption spectrum (**a**) and electron circular dichroism (ECD) spectrum (**b**) of PM567 dyes.

Figure 3 shows the electron hole-to-density isosurface of the four main excited states and their TDM diagram. We examined the isosurface graph and the color-filling matrix graph and found that the first excited state of the molecule exhibits strong local excitation characteristics, and the electron and hole density were concentrated on the pyrromethene, as shown in Figure 3a. However, the transition characteristics of $S_3$ were different. The intensity corresponding to $S_3$ is relatively low, and a strong charge transfer characteristic can be seen by looking at the electron-hole density map in Figure 3b. The electrons are transferred from dinaphthalene to pyrromethene. The weaker vibrator strength can also be confirmed from the side as the excited state of charge transfer. The other two main excited states $S_4$ and $S_6$ are localized excitations, as shown in Figure 3c,d. The difference is that the excited state of $S_4$ is the contribution of pyrromethene and the main contribution of $S_6$ is dinaphthalene.

The electron-hole density can show the transition characteristics, but only the overall transition dipole moment can be analyzed. During the interaction between molecules and electromagnetic waves, molecules respond differently to the electric and magnetic fields. In other words, different parts of the molecule respond differently to electric and magnetic fields. Therefore, molecules respond to polarized light to form an ECD spectrum. The ECD spectrum of PM567 dye is shown in Figure 2b. It can be seen that the ECD peaks contributed by $S_1$ and $S_2$ in the visible region show different positive and negative phenomena. Immediately afterward, $S_3$ and $S_4$ are also two oppositely excited states. Finally, $S_6$ is the strongest ECD peak in the positive direction.

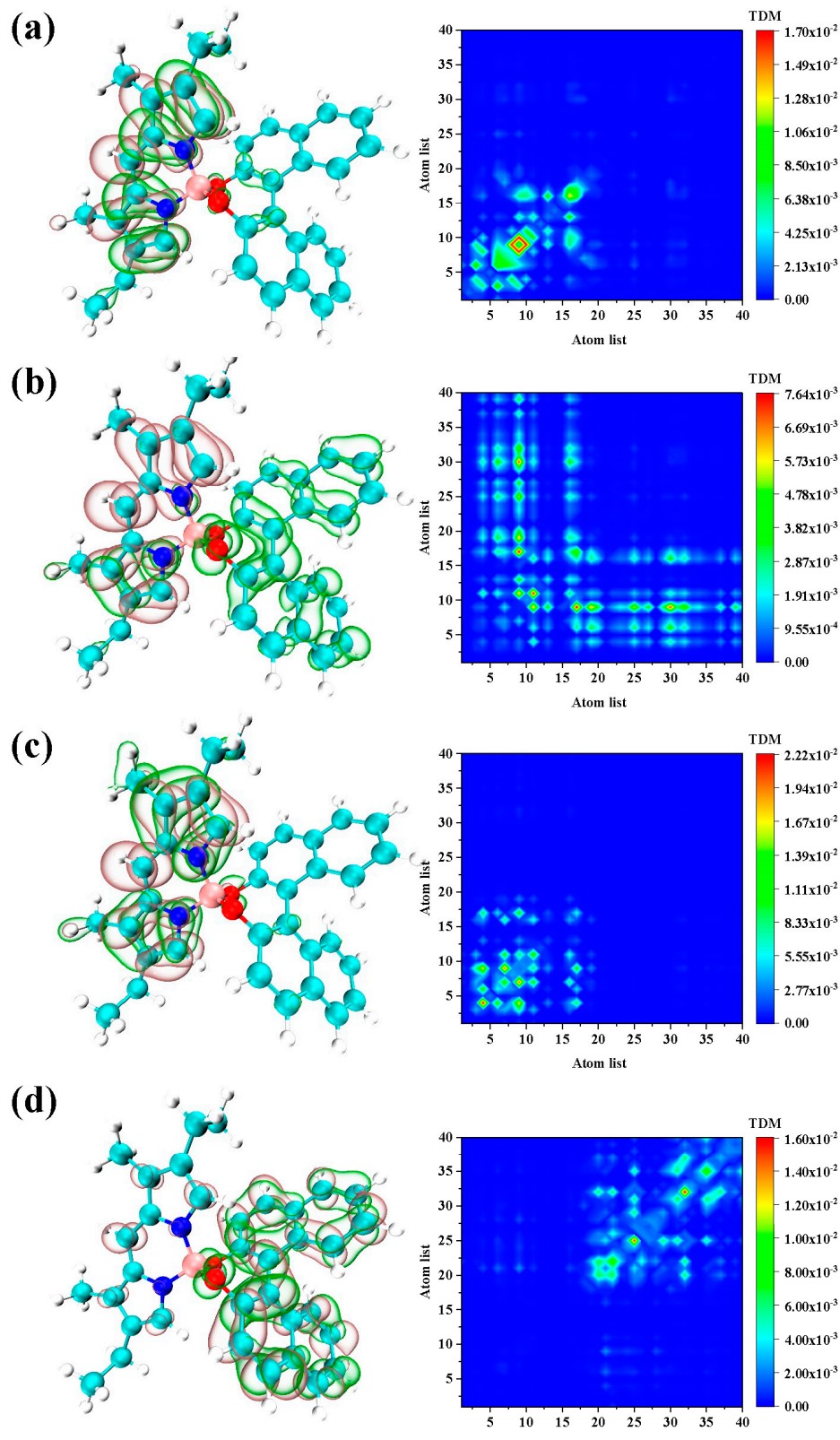

**Figure 3.** The electron-hole pair density (left) and transition density matrix (TDM) (right) of (**a**) $S_1$, (**b**) $S_3$, (**c**) $S_4$, and (**d**) $S_6$. The pink and green isosurfaces signify the electron and hole density, respectively.

To further analyze the mechanism of chiral electromagnetic interaction of each excited state, we decomposed and analyzed the electromagnetic interaction in the transition process. The transition electric dipole moment and transition magnetic dipole moment of $S_1$ were found to be distributed in the pyrromethene part. This showed that the $S_1$ excited state mainly contributes chirality by pyrromethene, as shown in Figure 4. This is an electromagnetic interaction different from the traditional chiral center. The ECD response of $S_2$ is the opposite to that of $S_1$, and the transition electric/magnetic dipole moment distribution is mainly distributed on pyrromethene. However, the dinaphthalene portion also has a large transition magnetic dipole moment density, as shown in Figure 5. Therefore, the chiral response of $S_2$ is determined by the entire molecule. The equivalent surface of the transition electric/magnetic dipole moment density of $S_2$ is relatively small, which is why the ECD peak corresponding to $S_2$ is relatively low. For $S_3$ and $S_4$, the difference between excitation energy and rotor strength is small.

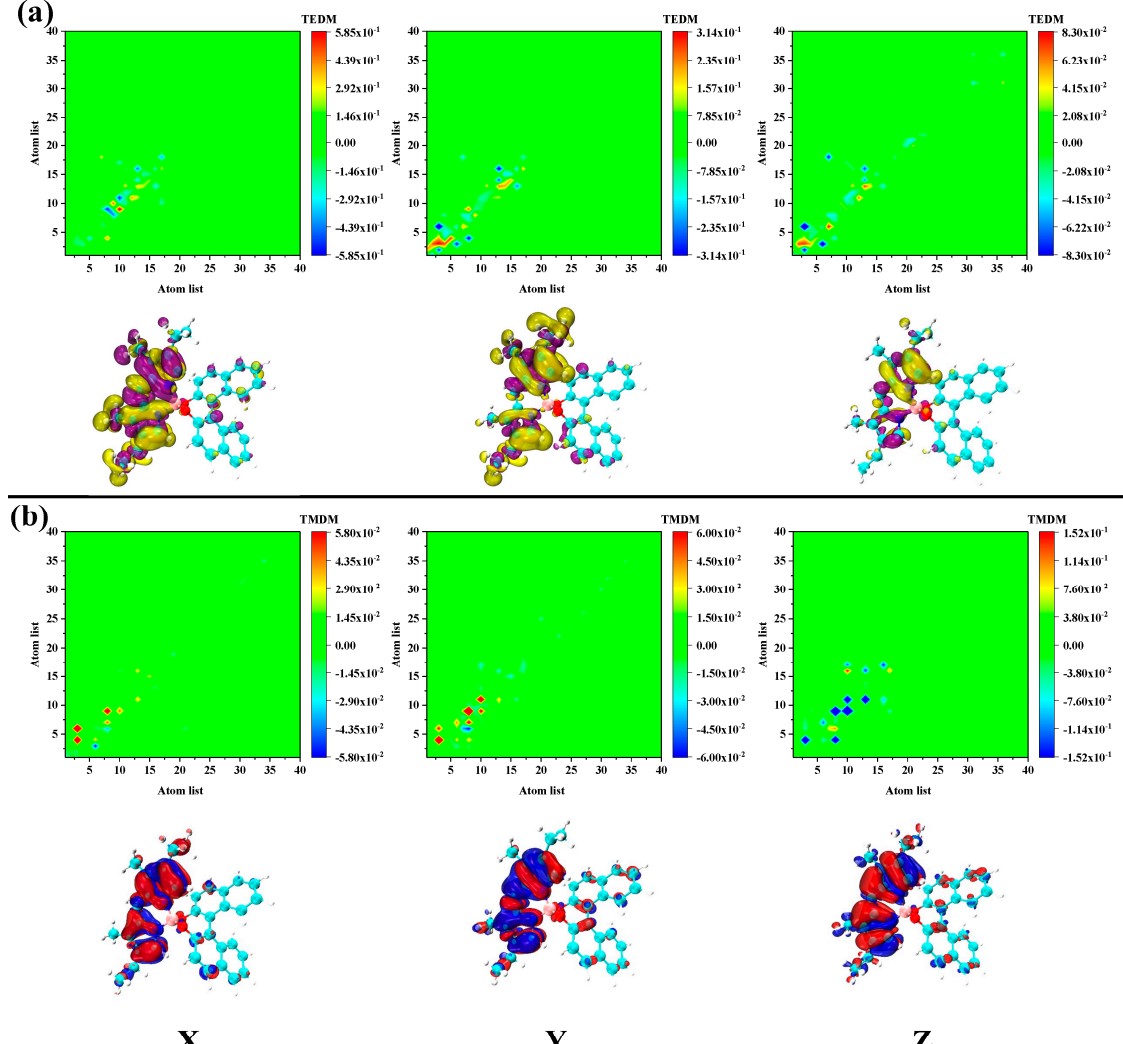

**Figure 4.** The transition electric (**a**) magnetic (**b**) dipole moment matrix (upper) and density (under) isosurface of $S_1$ in PM567. The yellow and purple isosurfaces represent the positive and negative transition electric dipole moments, respectively, and the red and blue isosurfaces denote the positive and negative transition magnetic dipole moments, respectively.

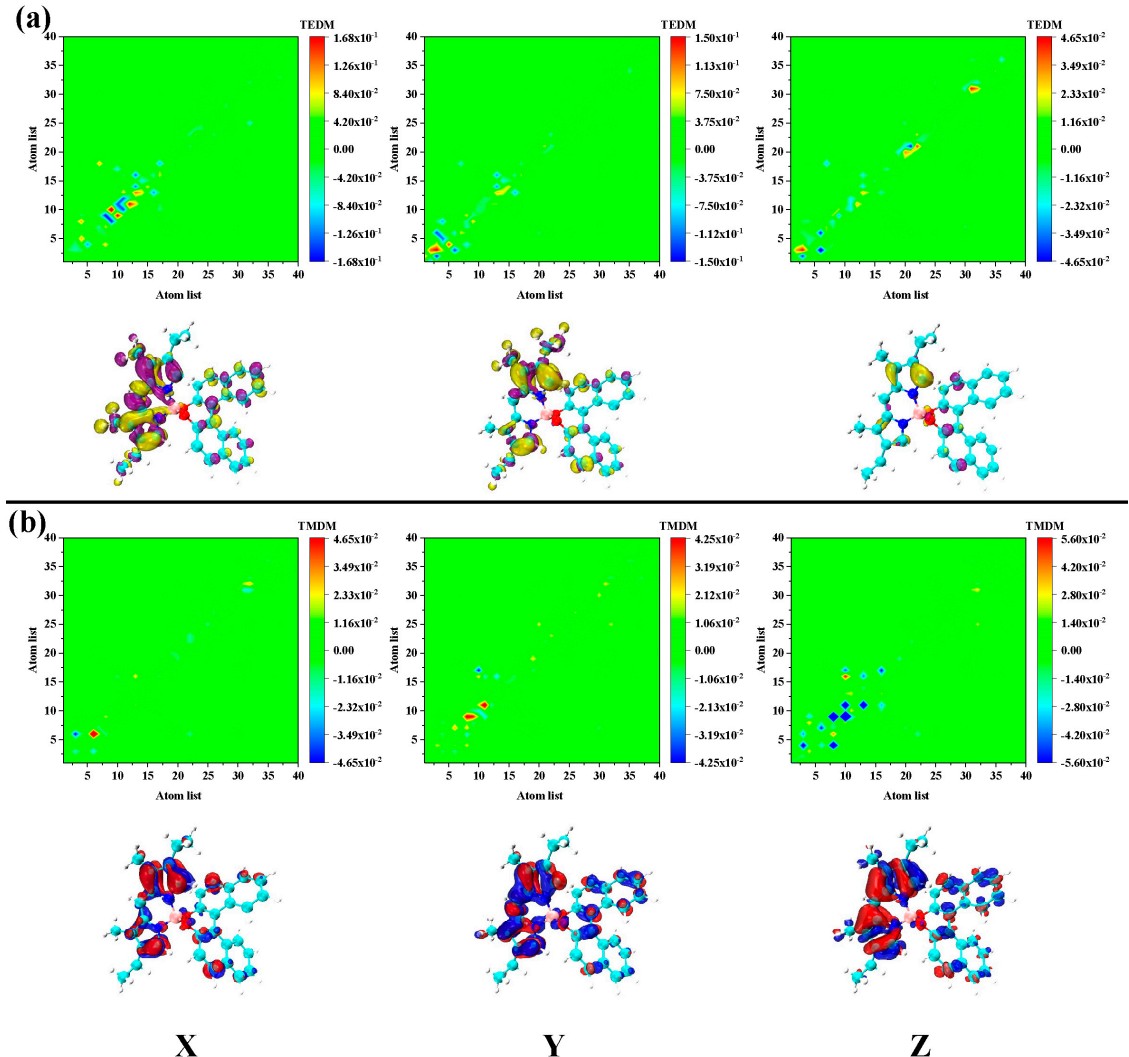

**Figure 5.** The transition (**a**) electric/magnetic (**b**) dipole moment matrix (upper) and density (under) isosurface of $S_2$ in PM567. The yellow and purple isosurfaces indicate the positive and negative transition electric dipole moments, respectively, and the red and blue isosurfaces represent the positive and negative transition magnetic dipole moment, respectively.

The transition electric/magnetic dipole moment deficiencies of these two excited states are significantly different. First, according to the previous statement, $S_3$ is the excited state of charge transfer. However, the density of $S_3$ transition magnetic dipole moment in the Y and Z directions is significantly higher than the electrical dipole moment density, as shown in Figure 6. The transition electric/magnetic dipole moment density distribution of $S_4$ is basically the same as $S_3$. However, the density distribution at the same location is the opposite, as shown in Figure 7. This is the fundamental reason why the ECD responses of $S_3$ and $S_4$ are opposing. The electromagnetic interaction of $S_4$ is significantly different from other excited states. Firstly, the transition electric/magnetic dipole moment density of $S_6$ is distributed in the binaphthalene part. Secondly, the transition electric/magnetic dipole moment density is exceptionally large, and significantly larger than the previous excited state, as shown in Figure 8. Correspondingly, the ECD peak of $S_6$ is also extraordinarily strong. Based on the above analysis, the chiral mechanism of the molecule was quantitatively analyzed after combining the transition electric/magnetic dipole moments (TEDM/TMDM) separately. The values of the transition electric/magnetic dipole moments of five different excited states are shown in Table 1. We calculated the tensor product between TEDM and TMDM according to Equation (2). The last row in Table 1 shows the eigenvalues of the tensor product, and the results exactly match the intensity and direction

of the ECD spectrum. This also showed that the analysis and discussion of the chiral mechanism mentioned above are complete and self-consistent.

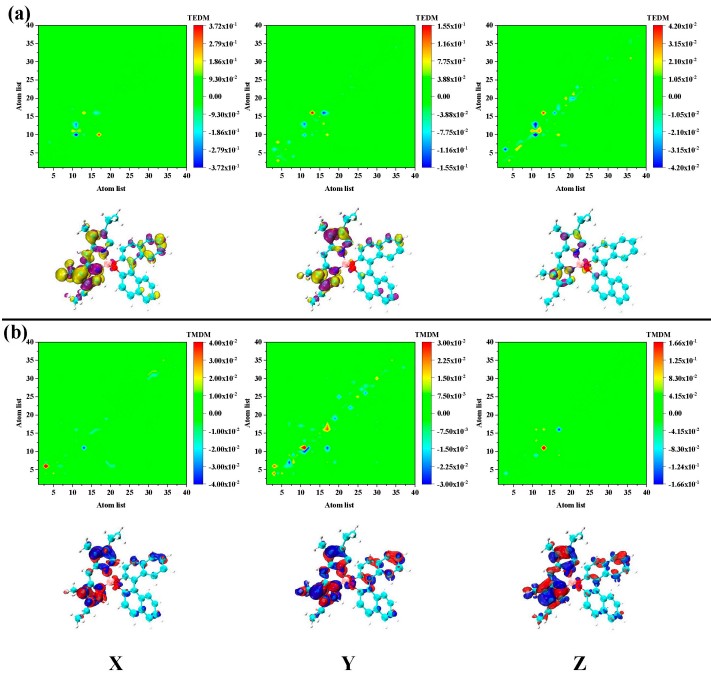

**Figure 6.** The transition (**a**) electric/magnetic (**b**) dipole moment matrix (upper) and density (under) isosurface of $S_3$ in PM567. The yellow and purple isosurfaces represent the positive and negative transition electric dipole moments, respectively, and the red and blue isosurfaces denote the positive and negative transition magnetic dipole moments, respectively.

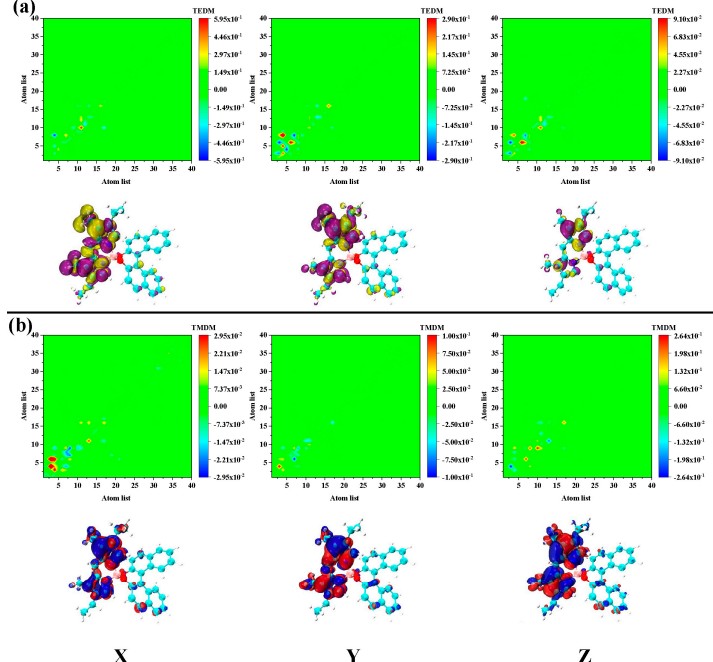

**Figure 7.** The transition (**a**) electric/magnetic (**b**) dipole moment matrix (upper) and density (under) isosurface of $S_4$ in PM567. The yellow and purple isosurfaces stand for the positive and negative transition electric dipole moments, respectively, and the red and blue isosurfaces stand for the positive and negative transition magnetic dipole moments, respectively.

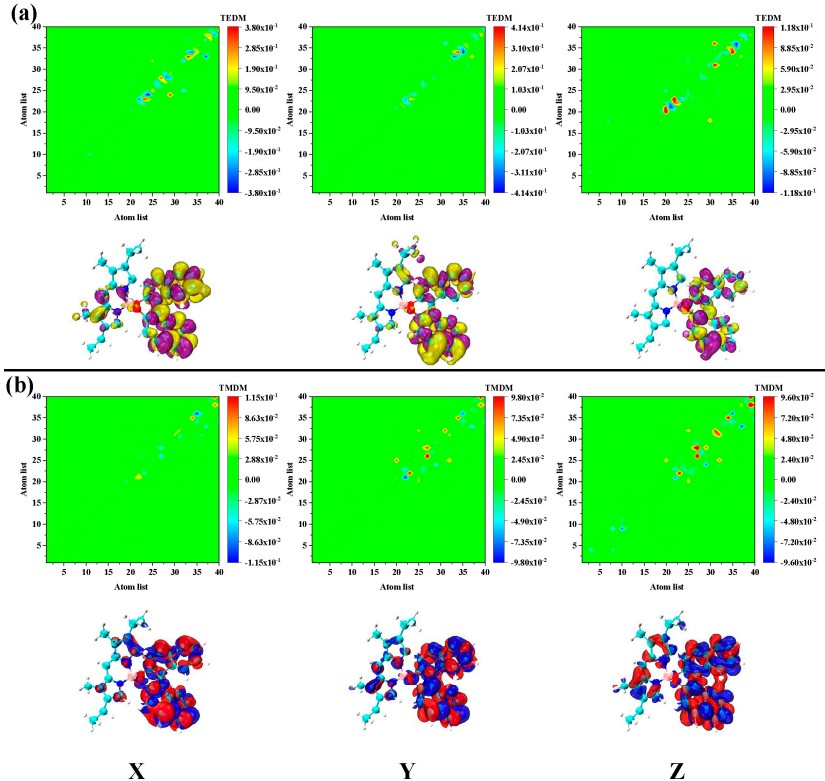

**Figure 8.** The transition (**a**) electric/magnetic (**b**) dipole moment matrix (upper) and density (under) isosurface of $S_6$ in PM567. The yellow and purple isosurfaces stand for the positive and negative transition electric dipole moments, respectively, and the red and blue isosurfaces stand for the positive and negative transition magnetic dipole moments, respectively.

**Table 1.** The value of transition electric/magnetic dipole moment and the eigenvalue of their tensor product.

|  |  | $S_1$ | $S_2$ | $S_3$ | $S_4$ | $S_6$ |
|---|---|---|---|---|---|---|
|  | X | −1.0509 | −0.2037 | −0.9342 | 0.2153 | −0.1139 |
| TEDM | Y | −2.0543 | −1.0207 | 0.0663 | 1.1107 | −1.4158 |
|  | Z | −0.5993 | −1.1083 | −0.0632 | 0.3309 | 0.4909 |
|  | X | 0.3253 | 0.0835 | 0.1045 | 0.247 | 0.266 |
| TMEM | Y | 0.3235 | 0.3412 | −0.0583 | 0.0245 | 0.7349 |
|  | Z | −2.8867 | −0.7827 | 0.089 | 0.3917 | 0.8694 |
|  | Eigenvalue | −0.7234 | 0.2327 | −0.0994 | −0.21 | 0.644 |

## 3. Method

Experimentally, electronic circular dichroism (ECD), vibration circular dichroism (VCD), and Raman spectroscopy (ROA) can be used to observe the chirality of molecules. ECD can effectively characterize the chirality of chromophores in molecules. In theory, the intensity of ECD can be defined as [26]:

$$I \propto \left\langle \varphi_j \middle| \mu_e \middle| \varphi_i \right\rangle \left\langle \varphi_j \middle| \mu_m \middle| \varphi_i \right\rangle B \tag{2}$$

where the $\varphi_i$ and $\varphi_j$ are the ground and excited state wave function, respectively. The B is the magnetic induction intensity. The $\mu_e$ and $\mu_m$ are the transition electric/magnetic moment, respectively. The transition electric dipole moment is defined by:

$$D_2^{(\mu)} = P_{\mu\mu}^{\tan} \left\langle \chi_\mu \middle| -z \middle| \chi_\mu \right\rangle + \sum_{v \neq \mu} \left[ P_{\mu v}^{\tan} \left\langle \chi_\mu \middle| -z \middle| \chi_v \right\rangle + P_{v\mu}^{\tan} \left\langle \chi_v \middle| -z \middle| \chi_\mu \right\rangle \right]/2 \tag{3}$$

where the $P_{\mu v}^{t \cdot cm} = \sum_{i}^{occ} \sum_{j}^{vir} w_K^{i \rightarrow j} C_{\mu i} C_{\mu j}$ is the transition density matrix, $C_{\mu i}$ and $C_{vi}$ are the linear combination coefficients of molecular orbitals, and $\mu$ is the number of basis functions. This allows calculation of the atomic contribution of the transition electric dipole moment. When analyzing ECD and ROA spectra, a transition magnetic dipole moment is also required, which is defined as follows:

$$M_z^{(\mu)} = P_{\mu\mu}^{\text{tran}} \left\langle \chi_\mu \middle| x\frac{\partial}{\partial y} - y\frac{\partial}{\partial x} \middle| \chi_\mu \right\rangle + \sum_{v \neq \mu} \left( P_{\mu v}^{\text{tan}} \left\langle \chi_\mu \middle| x\frac{\partial}{\partial y} - y\frac{\partial}{\partial x} \middle| \chi_v \right\rangle + P_{v\mu}^{\text{tan}} \left\langle \chi_v \middle| x\frac{\partial}{\partial y} - y\frac{\partial}{\partial x} \middle| \chi_\mu \right\rangle \right)/2 \quad (4)$$

On this basis, molecular structure optimization was achieved with Gaussian 16 software combined with the density functional theory (DFT) method [27], B3LYP functional [28], 6-31G(d) basis function, and the DFT-D3 [29] correction method. After optimizing the molecular structure, the TDDFT method was used in combination with the CAM-B3LYP functional [30] and the same basis function. After this, we combined it with Multiwfn-3.6 software [31], VMD-1.9.3. software [32], and self-programming [24] to visually analyze the chiral electromagnetic interaction.

## 4. Conclusions

In this work, we first performed a visual analysis of the absorption spectrum of the PM567 dye and the corresponding excitation characteristics of the excited state. We found that there are charge transfer excitation characteristics in the visible region and near ultraviolet region. This is an especially important conclusion for the photoredox reaction, which can be used to design photocatalytic reactions. Secondly, we conducted a decomposition analysis of the chiral electromagnetic interaction between molecules and electromagnetic waves during light excitation, explaining the source of the chiral spectrum. The chiral electromagnetic interactions in PM567 mainly come from the pyrromethene and binaphthalene moieties. This asymmetric chiral electromagnetic interaction is the key basis for the selection of the asymmetric catalytic reaction path driven by photon.

**Author Contributions:** Investigation, Y.D.; Resources, J.W. and N.F.; Data Curation, C.L.; Writing-Original Draft Preparation, J.W.; Data Curation, L.L. All authors have read and agreed to the published version of the manuscript.

**Funding:** This work was supported by the Fundamental Research Funds for the Central Universities, the Scientific Research Fund Project of Education Department of Liaoning province (No. L2019028); the Fundamental Research Funds for the Central Universities, and talent scientific research fund of LSHU (No. 2018XJJ-007); the National Nature Science Foundation of China under Grant 61901274; in part by the Shenzhen Science and Technology Innovation Committee under Grant JCYJ20190808141818890; and in part by the Guangdong Natural Science Foundation under Grant 2020A1515010467.

**Conflicts of Interest:** The authors declare no conflict of interest.

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
