# Peer review of "Visible Light Electromagnetic Interaction of PM567 Chiral Dye for Asymmetric Photocatalysis, a First-Principles Investigation"

_catalysts, doi:10.3390/catal10080882_

Round 1
Reviewer 1 Report
This reviewer believes that the manuscript can be accepted as it stands.
Author Response
This reviewer believes that the manuscript can be accepted as it stands.
Author reply: We are very grateful for the positive comments of the reviewers.
Reviewer 2 Report
The manuscript was well prepared. However, a detailed analysis leaves issues that should be addressed in the revised version of the manuscript.
1. The authors should focus on the maximum sensitivity of the studied PM567 dye. Typically for photocatalysis, longer irradiation wavelength (excitation source) are desired such as 450 nm. Based on Fig. 2 it is possible to excite dye with using longer irradiation wavelength.
2. What with the stability of the PM567 dye in long period time? This issue is important because dyes play a role of sensibiliser of semiconductors.
3. How the transfer of an electron from the dye to the semiconductor would take place?
4. The quality of the Figures is very poor - it should be improved. The introduction should present the use of the dye in photocatalysis (based on a detailed literature review).

Author Response
The manuscript was well prepared. However, a detailed analysis leaves issues that should be addressed in the revised version of the manuscript.
1. The authors should focus on the maximum sensitivity of the studied PM567 dye. Typically for photocatalysis, longer irradiation wavelength (excitation source) are desired such as 450 nm. Based on Fig. 2 it is possible to excite dye with using longer irradiation wavelength.
Author reply: We thanks the reviewer’s advice. The sensitivity of the absorption spectrum is positively related to the intensity of the oscillator. For the first excited state, 450nm light can indeed excite molecules. However, the absorption intensity is relatively small compared to the maximum peak. The sentence “In fact, the sensitivity of the absorption spectrum is positively related to the vibrator intensity. For the first excited state, longer wavelength light can indeed excite molecules. But compared to the maximum peak, the absorption intensity is relatively small, and the excitation efficiency is relatively low.” has been added and marked in red.
What with the stability of the PM567 dye in long period time? This issue is important because dyes play a role of sensibiliser of semiconductors.
Author reply: We thanks the reviewer’s advice. The stability of molecules after exposure to light is actually a problem of excited state dynamics. The TDDFT algorithm focuses on the steady-state problem. Therefore, the excited state dynamics is not suitable for the current theoretical system. However, the stability of molecules under light is based on geometric structure, which is a kind of excited state reaction problem. The main point of the article is the mechanism of the chiral interaction between molecules and electromagnetic waves in the excitation process, so this part is not studied in depth. The stability of PM567 has been discussed in references 19 and 20 in this article.
How the transfer of an electron from the dye to the semiconductor would take place?
Author reply: We thanks the reviewer’s advice. The calculation model involved in the article does not involve semiconductors.
The quality of the Figures is very poor - it should be improved. The introduction should present the use of the dye in photocatalysis (based on a detailed literature review).
Author reply: We thanks the reviewer’s advice. High-resolution versions of all pictures will be uploaded together with the revised comments. For the photocatalysis, the sentences “The chiral electromagnetic interaction between molecules and light is extremely important for asymmetric photocatalytic reactions. The exploration of the chiral electromagnetic interaction mechanism of dyes has guiding significance for the design of asymmetric photocatalytic reactions.” have been added in the section of introduction and marked in red.